# Universally autonomous self-healing elastomer with high stretchability

Hongshuang Guo[1,2,3], Yi Han[4], Weiqiang Zhao[1,2,3], Jing Yang [1,2,3 ✉] & Lei Zhang [1,2,3 ✉]

Developing autonomous self-healing materials for applications in harsh conditions is challenging because the reconstruction of interaction in material for self-healing will experience significant resistance and fail. Herein, a universally self-healing and highly stretchable supramolecular elastomer is designed by synergistically incorporating multi-strength H-bonds and disulfide metathesis in polydimethylsiloxane polymers. The resultant elastomer exhibits high stretchability for both unnotched (14000%) and notched (1300%) samples. It achieves fast autonomous self-healing under universal conditions, including at room temperature (10 min for healing), ultralow temperature ($-40\,^\circ$C), underwater (93% healing efficiency), supercooled high-concentrated saltwater (30% NaCl solution at $-10\,^\circ$C, 89% efficiency), and strong acid/alkali environment (pH = 0 or 14, 88% or 84% efficiency). These properties are attributable to synergistic interaction of the dynamic strong and weak H-bonds and stronger disulfide bonds. A self-healing and stretchable conducting device built with the developed elastomer is demonstrated, thereby providing a direction for future e-skin applications.

[1] Department of Biochemical Engineering, School of Chemical Engineering and Technology, Tianjin University, Tianjin 300350, P. R. China. [2] School of Chemical Engineering and Technology, Frontier Science Center for Synthetic Biology and Key Laboratory of Systems Bioengineering (MOE), Tianjin University, Tianjin 300350, P. R. China. [3] Qingdao Institute for Marine Technology of Tianjin University, Qingdao 266235, P. R. China. [4] Tianjin Key Laboratory of Molecular Optoelectronic Science, Department of Chemistry, Tianjin University, Tianjin 300350, P. R. China. ✉email: jing_yang@tju.edu.cn; lei_zhang@tju.edu.cn

Natural tissue, such as skin and muscle, possesses the ability to spontaneously heal injury, supporting the survival of most animals[1]. Recently, research is active in the development of synthetic materials that can mimic self-healing tissues for applications in e-skin, wearable electronic devices, and artificial muscles[2–9], in order to significantly improve the materials lifetime, robustness, and safety[10–20]. However, most designs of such healable materials require external energy (light, heat, or pressure) for healing[13,15,16,21–32], or the materials exhibit weak mechanical strength[22]. Overall, due to the different effects of strong or weak bonds in the materials, there is a compromise between dynamic healing and material strength.

Currently, dynamic supramolecular approaches are being developed to obtain tough self-healing polymers that allow autonomic healing of damage at room temperature without the need of external stimuli[21,33–40]. For example, Guan et al. designed a healable supramolecular elastomer based on non-covalent sacrificial bonds, merging the properties of toughness and autonomic healing under ambient conditions[17]. In addition, to achieve high stretchability and self-healing properties, Bao's group reported a chemical design of supramolecular polymer materials by using condensation polymerization, consisting of soft polymeric chains and reversible quadruple H-bonding crosslinkers[7]. However, for applications in various electronic equipment (e.g., electronic sensing device), harsh environments including marine area (supercooled seawater), polar regions (ultralow temperatures), or industrial wastewater (strong acid/alkali environment) cannot be avoided. Unfortunately, to date, the development of a universally healing material that is able to self-heal under all these harsh conditions still remains a great challenge. The reasons are mainly as follows: (i) When healable materials are injured or fractured underwater, water molecules can disturb the reconnection of dynamic bonds, such as the saturation of hydrogen bonds (H-bonds), the coordination with metal cations, or the solvation of ions, resulting in failure of the material to heal. (ii) Under freezing conditions, the dynamic character of bonds in healable materials can be significantly resisted, thus extremely restraining the self-healing process. Lower temperatures, more resistance. (iii) Some interactions for self-healing such as metal coordination bonds and dopamine are susceptible to pH changes.

In this work, we report a supramolecular elastomer design that merges the unique properties of high stretchability and universally autonomous self-healing. The key to this design is the synergistic interaction of multiple dynamic bonds including disulfide metathesis (S–S), strong crosslinking H-bonds (BNB–BNB), and weak crosslinking H-bonds (IP–IP, IP–BNB or IP–SS, etc.), as shown in Fig. 1. These dynamic bonds are introduced in a polydimethylsiloxane (PDMS) polymer backbone (PDMS–SS–IP–BNB), spontaneously forming a dynamic supramolecular polymer network (Fig. 1a and b). In the dynamic supramolecular polymer network, strong crosslinking H-bonds mainly impart robustness and elasticity to the elastomer as the weak H-bonds dissipate strain energy by efficient reversible bond rupture and reforming; the disulfide bonds mainly contribute to the self-healing property. The synergistic effect of these binding sites endows the elastomer with high stretchability (~14,000%) as well as fast autonomous self-healing ability at ambient condition, underwater, at ultralow temperature (−40 °C), in supercooled saltwater (30% NaCl solution at −10 °C), and even under strong acid/alkali environment (pH = 0 or 14) (Fig. 1c). Furthermore, we integrate the elastomer with eutectic gallium indium (EGaIn) (liquid metal) alloy, thus demonstrating the material application toward a stretchable self-healing artificial conductor.

## Results

A series of polymers were synthesized through a two-step polycondensation reaction. To synthesize a bis-isocyanate-terminated preoligomer, $\alpha$, $\omega$-dihydroxyethylpropoxyl-PDMS ($M_w = 4,600$ g mol$^{-1}$, as a soft segment diol) was reacted with isophorone diisocyanate (IPDI) monomer ($M_{(PDMS)}/M_{(IPDI)} = 1/2$) in N, N′-dimethylacetamide (DMAc) using dibutyltin dilaurate (DBTDL) as a catalyst. Then, 4,4′-dithiodianiline (SS) (an aromatic disulfide) and/or 4,4′-bis(hydroxymethyl)-2,2′-bipyridine (BNB) were added to the solution as a chain extender to complete the PDMS–SS–IP–BNB synthesis. PDMS–SS–IP–BNB polymers were thus linked by SS, BNB, and isophorone bisurea (IP) units of varying ratios (structures shown in Fig. 1 and Supplementary Fig. 1). Copolymers in Supplementary Table 1 are therefore identified by a shorthand name (P1–P7) that corresponds to the ratio.

We employed PDMS as a chain backbone, because it led to a more flexible and less crystalline polymeric backbone. In addition, PDMS has been reported to be nontoxic and is widely used as an ideal carrier material for e-skin devices and components. Based on PDMS, we designed multiple non-covalent crosslinkers, weak and strong H-bonds to mainly endow good mechanical properties to the polymer elastomer. Notably, the presence of SS and BNB units did not disrupt the PDMS-based soft domain, but instead induced PDMS–SS–IP–BNB to be soft as well as highly tough and stretchable, thus enabling it to self-heal in universal conditions.

**Material characterization.** ${}^1$H NMR indicated the successful synthesis of the PDMS–SS–IP–BNB polymers, as shown by the presence of characteristic peaks of PDMS, SS, IP, and BNB segments in the polymeric backbones (Supplementary Figs. 2 and 3). For P3, P4, P6, and P7, the Ar–H signals on the BNB units were very obvious when compared with that of P1 and P2 (Supplementary Fig. 3). Carrying out Fourier transform infrared spectroscopy (FTIR) and X-ray photoelectron spectroscopy (XPS) also proved the successful synthesis of all the PDMS–SS–IP–BNB polymers (Supplementary Figs. 4 and 5). For example, the FTIR spectra of all polymers featured negligible peaks at 2264 cm$^{-1}$, which corresponded to N=C=O stretching. It indicated that the diisocyanate monomers (IP) were fully converted to urethane bonds. Furthermore, characteristic peaks of PDMS–SS–IP–BNB appeared at 1705 and 1547 cm$^{-1}$, which corresponded to C=O stretching and N–H bending, respectively. The comparable number-averaged molecular weight ($M_n$) of the polymers was obtained by gel permeation chromatography (GPC) using THF as the eluent and poly (methyl methacrylate) (PMMA) as the standard. $M_n$ was in the range of 22–66 kDa and exhibited relatively narrow polydispersity indices (PDI ≤ 1.79, calculated by $M_w/M_n$) (Supplementary Table 2). The polymers featured excellent thermal stability with a decomposition temperature over 290 °C (Supplementary Fig. 6). In addition, using differential scanning calorimetry (DSC), the glass transition temperature ($T_g$) of the polymeric backbones of P3, P4, and P7 were all below −50 °C, which was highly important for achieving low modulus, toughness, and self-healing ability in the PDMS–SS–IP–BNB elastomer under ultralow temperatures (Supplementary Fig. 7)

**Rheological and mechanical properties.** Rheological measurements can reveal the viscosity and elasticity of materials and thus enable explanation of their mechanical properties (Supplementary Fig. 8). The results at room temperature in Supplementary Fig. 8a–e show that the storage modulus (G′) of all PDMS–SS–IP–BNB films (P1, P3, P4, P5, and P7) was greater than the loss modulus (G″), presenting a characteristic of elastomer. With increasing frequency, G′ increased significantly faster

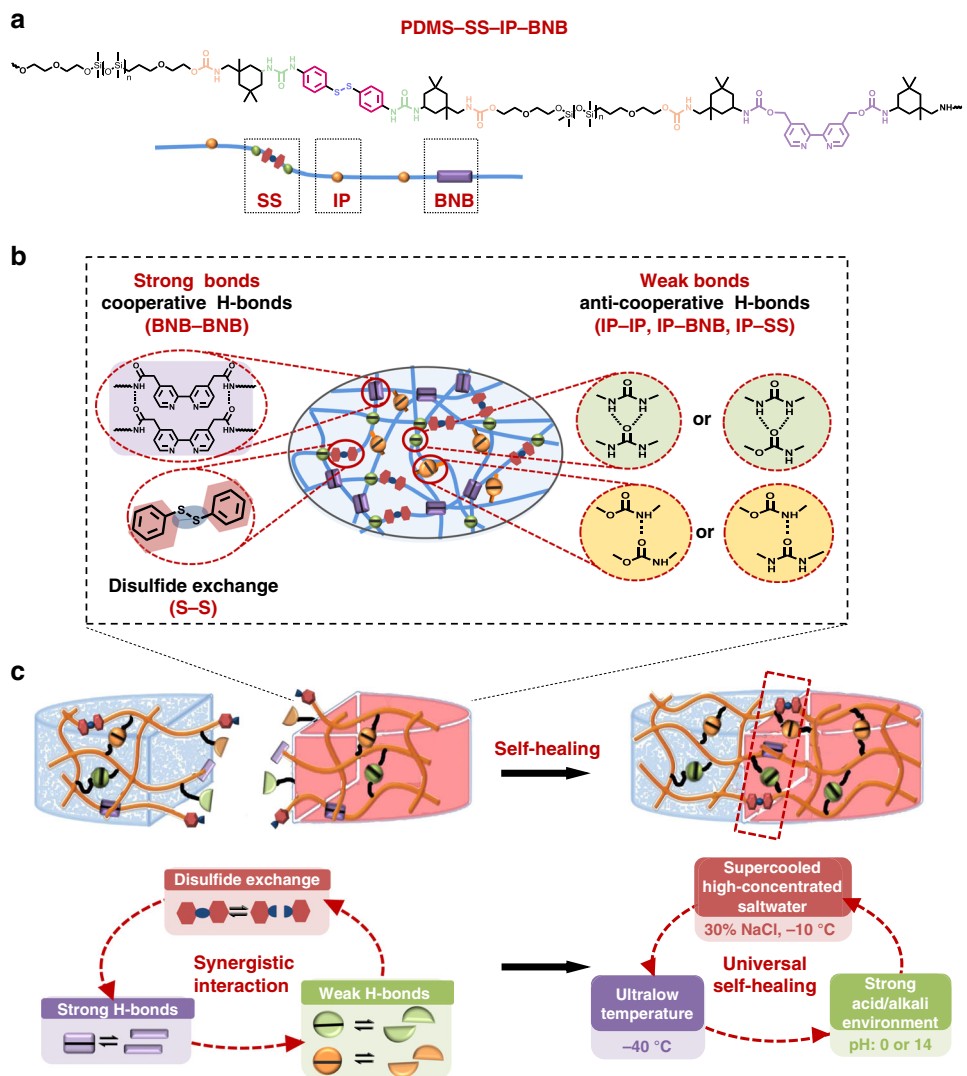

**Fig. 1 Molecular design of the PDMS–SS–IP–BNB elastomer with high toughness, stretchability, and universally autonomous self-healing ability. a** Chemical structure of PDMS–SS–IP–BNB. **b** The proposed ideal structure of the supramolecular polymer network based on strong crosslinking H-bonds (BNB–BNB), weak crosslinking H-bonds (IP–IP, IP–BNB, IP–SS), and disulfide metathesis (S–S). **c** The synergistic interaction of multiple dynamic bonds contributes to universally self-healing capability of elastomer.

than G″, indicating that PDMS–SS–IP–BNB film showed elasticity rather than viscosity and mainly elastic deformation. But when the temperature is higher than 60 °C in Supplementary Fig. 8g–j, G″ became greater than G′ suggesting the enhanced mobility of polymer upon heating and a solid-liquid transition to viscous deformation. These results indicated that PDMS–SS–IP–BNB materials mainly possessed an elastic property at room temperature and could become more viscous and fluid-like at higher temperatures. This is also consistent with the lower $T_g$ of the materials (Supplementary Fig. 7).

Figure 2 shows that the resulting PDMS–SS–IP–BNB film possessed a very high stretchability and good mechanical strength. Based on a reasonable ratio design, the film (P4) could be surprisingly stretched to nearly 140 times its original length at a loading rate of 10 mm min$^{-1}$ without rupture (Fig. 2a, b). Moreover, the P4 film also maintained a high stretchability even stretched much faster (Supplementary Fig. 9). To investigate the origin of the remarkable mechanical properties of PDMS–SS–IP–BNB, different unit ratios of PDMS–SS–IP–BNB and their parent polymers (P1, P2: PDMS–SS–IP; P5: PDMS–IP–BNB) were synthesized. As shown in Fig. 2b and c,

the stress–strain curves of the polymer films all included an initial stiffening region (tension stress was proportional to strain), and a subsequent steady region (stress was almost constant) until rupture. It is a typical stress–strain curve of elastomer in the presence of interchain momentary bonds. P2 presents a lower stretchability (100%) in Fig. 2b; the lack of BNB units allowed for strong H-bonds of BNB–BNB crosslinks. The strain at break of the P5 film (lack of SS) was ca. 2,000%, higher than that of P2 but still much lower than that of P4 (14,000%). This result is also consistent with the lowest Young modulus (P4) among the polymers (Supplementary Table 2). We suggest that multiple non-covalent and covalent interactions were synergistically improve the material mechanical strength; furthermore, the strong H-bonds (BNB–BNB) played the most important role in high stretchability of PDMS–SS–IP–BNB.

When the PDMS content was reduced, the film (P6) lost its elasticity and underwent continuous plastic deformation upon applying strain. We consider that this is a result of PDMS being a linear polymer with good flexibility, indicating the contribution of PDMS soft segments to the tensile property of PDMS–SS–IP–BNB material. Figure 2c shows that the

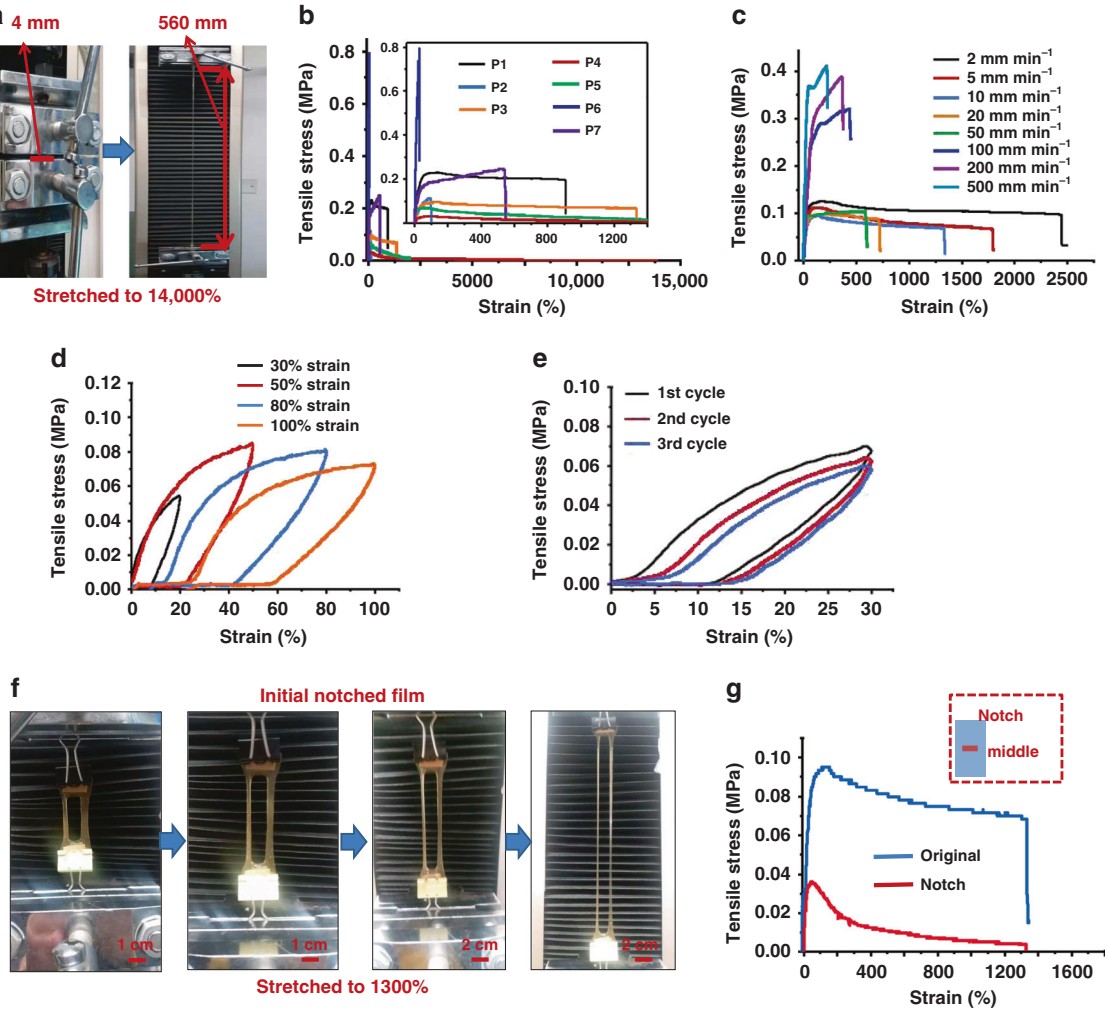

**Fig. 2 Mechanical properties of PDMS–SS–IP–BNB elastomer. a** Photographs of a PDMS–SS–IP–BNB elastomer film (P4) before and upon stretching. **b** Stress–strain curves of PDMS–SS–IP–BNB elastomer films prepared with different molar ratios. **c** Stress–strain curve of the PDMS–SS–IP–BNB elastomer film (P3) at different speeds. Sample width, 14 mm; thickness, 1 mm; gage length, 2 mm. **d** Stress–strain curves of the film (P3) in stress–strain tests (20%, 50%, 80%, 100% strain) in successive stretching. **e** The stress–strain curve of the film (P3) in cyclic stress–strain tests (30% strain) in successive stretching. Sample width, 14 mm; thickness, 1 mm; gage length, 40 mm. Stretching speed, 10 mm min$^{-1}$. **f, g** A notch-insensitive and highly stretchable elastomer film (P3). A notch (8 × 8 mm$^2$) with 2 mm thickness, 20 mm length, and 16 mm width (left) was made in the P3 film, and the film was stretched at a loading rate of 10 mm min$^{-1}$. The inset graph is the schematic diagram of notch location. Source data are provided as a Source Data file.

stretchability of PDMS–IP–SS–BNB film was strongly dependent on the stretching speed. Because of the increased strain speed, there was less time available for broken dynamic bonds to reform, significantly decreasing the fracture tolerance. The extension and recovery properties are presented in cyclic stress–strain tests (Fig. 2d, e and Supplementary Fig. 10). The curves showed a pronounced hysteresis, even for a lower applied strain (30%), which indicated strain energy dissipation during the rupture of the dynamic bonds in the stretching process. After the mechanical strain was removed, partially broken bonds could be recovered, and the mechanical properties of film were restored. These observations are similar to that in previously reported hydrogen-bonded crosslinked and ionic crosslinked elastomers[4,35]. In such materials, to dissipate the strain energy, non-covalent bonds break; after releasing the mechanical strain, most of the mechanical properties can be restored by recovering partially broken bonds. Moreover, the aromatic disulfide metathesis in PDMS–SS–IP–BNB can act as the sacrificial bonds for breakage to enhance the material strechability.

In e-skin or e-wear applications, the commonly used substrates usually suffer from instability, low toughness, or high notch sensitivity, which significantly reduces their service life. For example, typical PDMS (Sylgard 184) can be broken at 200% strain, polyurethane at 700% strain, and styrene butadiene styrene block copolymer (saturated) breaks at 280% strain[4]. When notched, all these materials achieve only <150% stretching strain. In this work, PDMS–IP–SS–BNB elastomer film exhibited excellent long-term stability (at least 5 months) under ambient conditions (Supplementary Fig. 11) and notch-insensitive ability for both middle and side notch (Fig. 2f, g and Supplementary Fig. 12), superior to that of most reported stretchable materials. Notably, when a big notch (7–8 mm) was introduced at the middle of the film (16 mm width), the film could achieve notch-insensitive stretching up to 1300% strain, which is similar to that of the original P3 film (Fig. 2f, g). This suggests that the notch in the film was blunted without tearing further during the stretching process, thus evidencing the extraordinary toughness of the PDMS–IP–SS–BNB material.

Based on our results, we can propose that the high stretchability of PDMS–SS–IP–BNB material is attributed to synergistic interaction of multiple dynamic bonding features in the supramolecular polymer network. The mechanical properties of

the material depend on the ratio of the PDMS, SS, BNB, and IP units, notably, a higher BNB–BNB crosslinking (strong H-bonding) density yields higher mechanical strength. In the supramolecular polymer network, the strong H-bonding (quadruple H-bonding) between BNB formed in a cooperative manner[41], thereby allowing the much stronger BNB–BNB crosslinking to hold the elastomer together for elasticity (Fig. 1b). On the contrary, due to the steric hindrance caused by the isophorone moieties, IP units in the polymeric backbones can only form maximum dual H-bonding with another IP, SS, or BNB units, resulting in weak H-bonding for efficient energy dissipation in the supramolecular polymer network[41,42]. As shown in Supplementary Fig. 13, the strong and weak H-bonds present intrachain and interchain H-bond interactions. Intrachain H-bond interactions lead to the folding of the PDMS–SS–IP–BNB chain, allowing for a large chain extension; whereas the interchain H-bond interactions result in a 3D crosslinking network and are repeatedly bonded/broken during the sliding between chains[43]. In addition, the breakage of covalent bonds (S–S) can dissipate the energy under tension, which also significantly benefited the toughness of the material. Therefore, the synergistic effect of three kinds of dynamic bond sites in polymeric backbones provided both strong crosslinking interaction and an energy dissipation mechanism, offering multiple modes of bond breaking, exchanging, and reforming. It could be advantageous for mechanical strength of the resulting PDMS–IP–SS–BNB material. When a notched PDMS–SS–IP–BNB material is stretched, the strong H-bonds efficiently prevent the strain-induced crack propagation; simultaneously, the weak H-bonds and S–S bonds break to dissipate strain energy. We assume that breaking the weak H-bonds and S–S bonds reduces the stress concentration at the notch.

**Universally self-healing capability**. During the past decade, much effort has been made to develop self-healing materials that do not require external energy, such as light, heat, or additives[13–16,55–58]. However, a few materials achieved autonomous self-healing at ambient conditions[16,17], and it remains challenging to realize autonomous self-healing in harsh conditions, such as underwater, at freezing or even ultralow temperatures, in supercooled seawater, or in strong acid/alkali conditions. To the best of our knowledge, there is no currently reported material that can achieve universally self-healing in all above mentioned harsh conditions. In this work, PDMS–SS–IP–BNB elastomer not only possesses a high stretchability but also exhibits universally self-healing capability with high healing efficiency. The elastomer was cut into two pieces that were subsequently put together for healing without any external simulation under different periods and conditions (Fig. 3 and Supplementary Figs. 14–19).

After 1-h healing at room temperature, the trace of cut was almost invisible (Supplementary Fig. 14). One of the two pieces was dyed red to make the cut area more discernible. As shown in Fig. 3a, after a healing duration of 10 min at room temperature, the healed P3 film sustained a large strain and held a weight that was 526 times greater than the film weight. As expected, a longer healing time resulted in a higher recovered fracture strain (Supplementary Fig. 16a and b), When healed at room temperature for 2 h, the film had a recovered fracture strain of 1200% ± 50%, which showed a high healing efficiency ($\eta$) of 93% ± 3% (Supplementary Fig. 16b). Meanwhile, the P4 film also presented quick and efficient self-healing ability in ambient conditions, as shown in Supplementary Fig. 15. Compared to the previously reported self-healing elastomers and hydrogels reported, PDMS–SS–IP–BNB represented a significant advance in self-healing time and strechability (Supplementary Fig. 19). In contrast, parent films (P1 and P5) presented the significantly

lower ambient self-healing efficiency in comparison to that of P3 film, and notably the P5 film that was lack of S–S bonds performed worst. It suggested that the excellent ambient self-healing ability of the PDMS–SS–IP–BNB elastomer was attributed to synergistic interaction of multiple dynamic bonds in the supramolecular polymer network, with the S–S bond being essential.

In addition to healing at damaged locations, the film demonstrates connection by self-healing (Supplementary Fig. 17a); two undamaged PDMS–SS–IP–BNB (P3) films were cut into two pieces, and the undamaged edges were placed next to each other. After healing at room temperature for 2 h, the rejoined film sustained a 92.9% ± 30% strain with a 68% ± 3% healing efficiency (Supplementary Fig. 17b). When compared with a cut edge with a higher healing efficiency (93% ± 3%), there were fewer reactive sites available on the unbroken and rough edges, inducing an insufficient molecular diffusion. Likely, the dynamic exchange of disulfide metathesis and multi-strength H-bonds were mainly present in the bulk of the polymer matrix rather than on the surface. In comparison with other self-healing materials[16,17,57,59,60–62], PDMS–SS–IP–BNB elastomer possesses another superiority because its healing ability is not significantly affected by surface aging. P3 film was cut into two pieces and healed after a 24-h separation in ambient conditions; the healing still reaches 68% ± 2% efficiency (Fig. 3d). Moreover, the cycle of stretching, breaking, and healing of the film could be repeated multiple times, demonstrating its superior stability.

Furthermore, we evaluated the universally self-healing properties in various harsh conditions, including underwater, at refrigerated (4 °C), freezing (−20 °C), and ultralow (−40 °C) temperatures, in a supercooled saline solution (30% NaCl solution at −10 °C), and in strong acid/alkali environment (pH = 0 or 14). Unlike other H-bond based self-healing materials (such as hydrogels)[16,17], PDMS–SS–IP–BNB film presented a water-insensitive property and achieved autonomic healing underwater (Supplementary Fig. 18, Fig. 3d). When the severed P3 film was healed in water for 24 h, the resulting film could still be stretched up to 1230% ± 30% strain with 93% ± 2% healing efficiency. We assume that the hydrophobicity of the polymer backbone (PDMS) gives rise to water-insensitive self-healing. The hydrophobic property induces an ordered arrangement of the surrounding water molecules, thus significantly reducing entropy such that the hydration of H-bond units in the elastomer is avoided. In other self-healing materials, the hydration of dynamic bonding units results in underwater self-healing failure, because water is a strong competitor for H-bonds, which can cause the dissociation of H-bonds. To verify this assumption, the film weight before and after immersion in water was recorded, and there was no significant water uptake into the film during immersion (Supplementary Fig. 18). The immersion in water (up to 24 h) led to no obvious reduction in the mechanical properties (Fig. 3d)[4,19,35].

In addition, the healing took place under refrigerated conditions (4 °C) with a high efficiency of 86% ± 2% and freezing conditions (−20 °C) with 55% ± 3% healing efficiency after 24 h, even 52% ± 4% healing efficiency at −40 °C after 24 h (Fig. 3d, e and Supplementary Video 1). This is the lowest temperature to realize autonomous self-healing that has been currently reported. One of the reasons for this phenomenon should be the viscous and fluid-like properties with a low $T_g$ (<−50 °C) of this elastomer, preventing it from freezing state and benefiting its self-healing efficacy due to the sufficient re-entanglement of polymer chains at the damaged surface even at ultralow temperatures (Supplementary Fig. 7)[36]. In addition, the sufficient multiple dynamic bonds can synergistically provide more

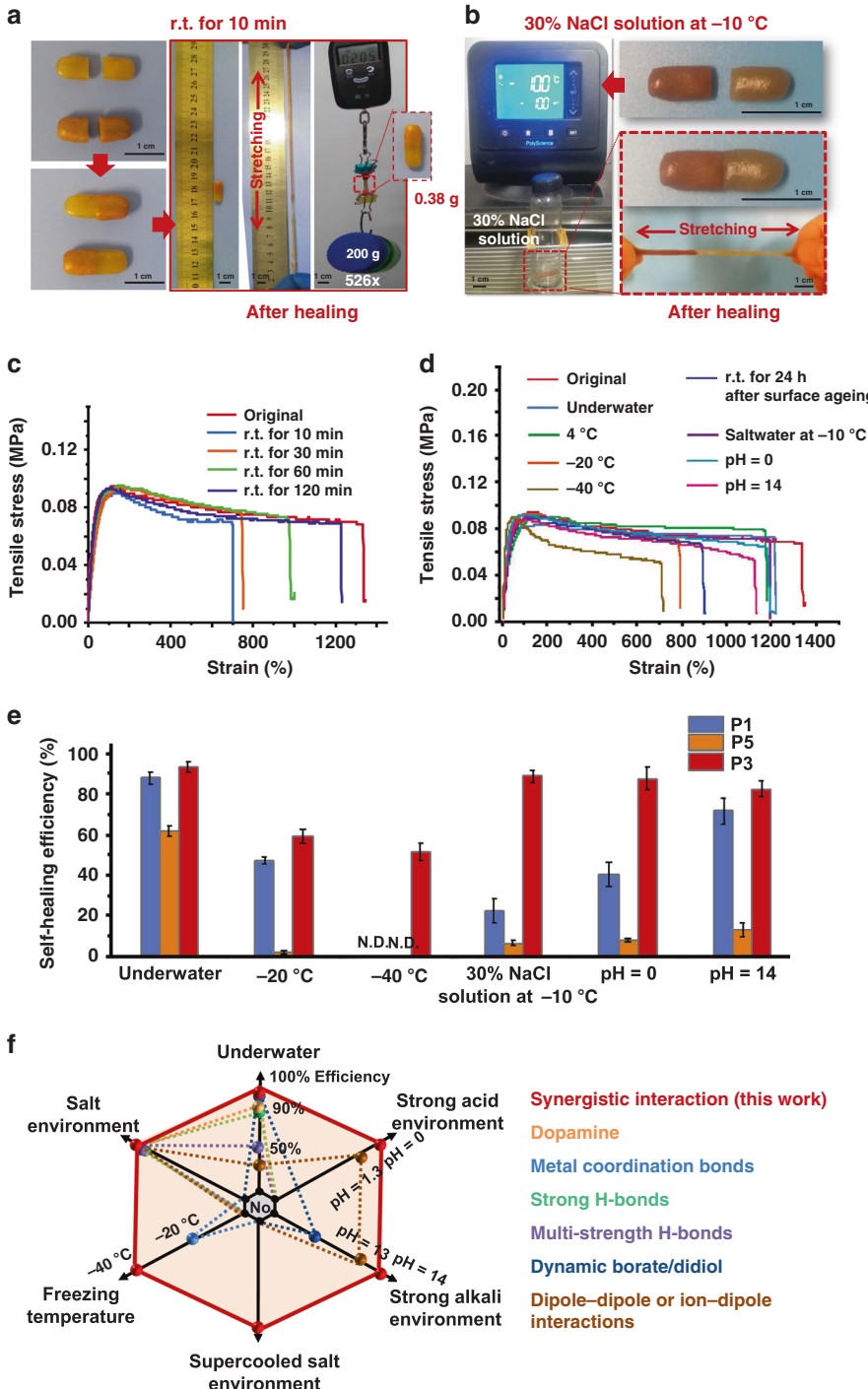

**Fig. 3 Universally self-healing capability of PDMS–SS–IP–BNB elastomer. a** Photographs of P3 film (left) before and (right) after self-healing and enabling high stretchability and holding a weight that is 526 times greater than that of the film. **b** Self-healing of P3 film in a 30% NaCl solution at −10 °C. P3 film is bisected and put together in a 30% NaCl solution at −10 °C. After 24 h self-healing, the film is removed and can be stretchable. **c** Stress–strain curves of the P3 film healed for different time periods at room temperature (r.t.). Stretching ability increases when the film is allowed to heal for a longer time. **d** The stress–strain curves of P3 film healed under universal conditions. Sample width, 14 mm; thickness, 1 mm; gage length, 2 mm. Stretching speed, 10 mm min⁻¹ **e** The self-healing efficiency (%) of P1 (lack of strong H-bonds), P3, and P5 (lack of S–S bonds) film healed underwater for 12 h, at −20 °C for 12 h, at −40 °C for 24 h, 30% NaCl solution at −10 °C for 24 h, pH = 0 and pH = 14 solution for 24 h. Sample size: 60 × 20 × 1.0 mm³; Stretching speed: 10 mm min⁻¹. Error bars = standard deviation (n = 3) **f** A comparison between this work and previously reported self-healing materials in terms of self-healing ability in universal harsh conditions (refs. [4,5,7,8,12,19,35,40,44-54]). Source data are provided as a Source Data file.

reconfiguration opportunities when polymer chains re-entangle at the damaged surface. More importantly, the low bond energy of disulfide metathesis can further ensure the easy reconnection at freezing conditions. To investigate a harsher condition, the film was placed in a 30% NaCl solution at −10 °C, as shown in Fig. 3b. The self-healing efficiency could reach up to 89% ± 3% after 24 h (Fig. 3d, e). Furthermore, even in pH = 0 or 14 solution, the healing efficiency could reach more than 83% ± 4% after 24 h (Fig. 3d, e and Supplementary Video 2). This elastomer system possessed an unparalleled self-healing ability in water/saltwater at such a low temperature without external stimulus, presenting an excellent stability in strong acid/alkali environment at the same time (Fig. 3f). It would be highly important for applications in self-healing electronic equipment in marine environment, polar regions, or industrial wastewater.

Figure 3e shows the self-healing efficiency of P1 (lack of strong H-bonds) and P5 (lack of S−S bonds) films in all of harsh conditions, to compare with P3 film based on synergistic interaction among three types of dynamic bonds. P1 showed a significant lower healing efficiency than P3, in particular the inability of anti-freezing self-healing at −20 °C and −40 °C. Meanwhile, negligible healing capability of P5 in the absence of S−S bonds was presented in all harsh conditions except underwater (only ~60% efficiency). As compared to most covalent bonds, the lower S−S bond dissociation energy (60 kcal mol⁻¹) and disulfide metathesis reaction efficiently improved the healing process and shortened the self-healing time[25,62]. Therefore, dynamic S−S bonds played the most important role in universally self-healing ability of PDMS−SS−IP−BNB elastomer. Notably, it could be observed that the healing efficiencies of P3 were significantly greater than the simple sum efficiencies of P1 and P5 based on two types of dynamic bonds. It suggested that the synergistic interaction among three types of dynamic bonds was a main reason for universally self-healing ability.

Therefore, the PDMS−SS−IP−BNB elastomer showed universally self-healing properties with high efficiency, which are attributed to these factors: (1) the synergistic interaction among the dynamic strong H-bonds, weak H-bonds, and disulfide metathesis bonds in a supramolecular polymer network, and the latest dynamic bonds playing the most important role in self-healing; (2) both the hydrophobicity and low $T_g$ of polymer backbone.

**A self-healing and stretchable conducting device.** The PDMS−SS−IP−BNB elastomer possesses exceptional mechanical, stretchable, and universally self-healing properties, which are highly favorable for a conductor applied in conductive device. In previous reports, using EGaIn for self-healing electrode materials was promising on account of its nontoxic and conducting properties[63]. Therefore, we fabricated a stretchable and autonomous self-healing conductor using the liquid metal EGaIn as a conducting wire and the P3 elastomer as the encapsulation and supporting layer (Fig. 4a)[64–66]. The detailed preparation process is presented in the Method. As shown in Fig. 4b (the upper row) and Supplementary Video 3, the LED was always on, which means that the conducting wire maintained electrical conductivity despite being stretched to 400% strain and recovered to its original state. In Fig. 4b (the lower row), when the conducting wire was cut in half, the circuit was open, leading to the LED being off. When the two parts were reconnected together, the electrical conductivity was restored immediately, whereas the mechanical properties required 10 min to recover. Liquid EGaIn aggregated and fused quickly to recover the electrical conductivity but slower dynamic bonding was necessary for mechanical healing. Moreover, the healed conducting wire maintained the electrical conductivity when stretched to 150%.

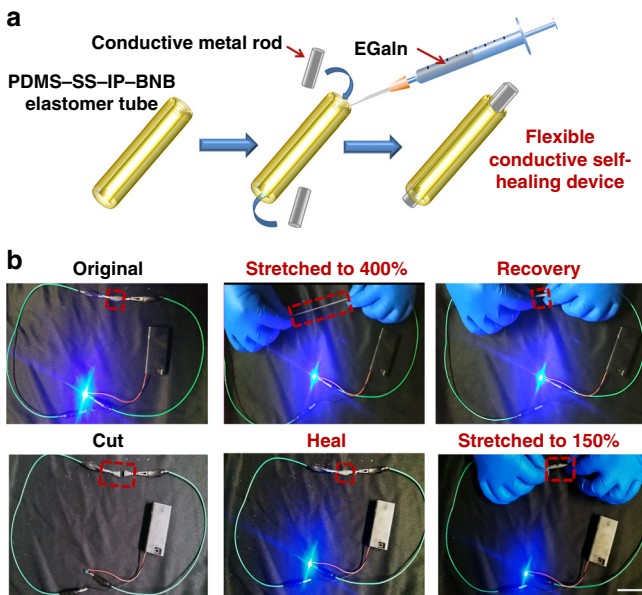

**Fig. 4 A self-healable and stretchable artificial conducting device based on PDMS-SS-IP-BNB elastomer. a** Components and structure of the electrical conductor using an elastomer tube. EGaIn is covered with the elastomer that serves as insulator. **b** The self-healable and stretchable conductor applied with an LED in series. The electrical conductor can be stretched to 400% its original length and recover its original size; moreover, the electrical conductivity of a broken conductor can be recovered after 10 min healing, and the healed conductor can still be stretched to 150% its original size (scale bar = 4.0 cm).

## Discussion

We have developed a universally autonomous self-healing and highly stretchable elastomer that synergistically incorporates multiple dynamic bonds as crosslinks for a supramolecular polymer network. The dynamic bonds include strong H-bonding interaction (most critical for stretchability), weak H-bonding interaction, and disulfide metathesis (most essential for self-healing), and their dynamic nature allows them to be broken and reformed, thus rendering an elastomer with excellent stretchability and self-healing properties. Notably, the elastomer exhibited universal self-healing capabilities in ambient conditions and harsh conditions including underwater, at freezing or even ultralow temperatures (−20 °C or −40 °C), in supercooled salt-water, or under strong acid/alkali conditions (pH = 0 or 14) without external stimulus or additives. This material's ability to autonomously heal itself in universal conditions makes it promising for a wide range of applications, such as artificial e-skin, and may enable the self-healing of electronic communication equipment involved in marine warfare and polar regions.

## Methods

**General considerations.** All the used reagents, starting materials and instrumentation descriptions are described in Supplementary Methods. Supplementary Video 1 is P3 self-healing at −40 °C for 24 h. Supplementary Video 2 is P3 self-healing in pH = 0 or 14 solution for 12 h. Supplementary Video 3 is a self-healable and stretchable artificial conducting device based on P3.

**Polymer synthesis.** A typical polymerization procedure for P3 is described below. HO−PDMS−OH (23.0 g, 5.0 mmol) in a dried glass vessel equipped with a mechanical stirrer was heated in an oil bath at 100 °C under vacuum (<133 Pa) for 1 h to remove any moisture, and then cooled to 70 °C. IPDI (2.44 g, 11 mmol) and DBTDL (0.08 g) were dissolved in DMAc (20 mL), dropwise added into the vessel, and stirred for 3 h under a N₂ atmosphere. After the synthesis of the pre-polymer, 4,4′-dithiodianiline (0.6 g, 2.5 mmol) and 4, 4′-bis(hydroxymethyl)-2,2′-bipyridine (0.465 g, 2.5 mmol) in 6 mL of dry DMAc were added to the pre-polymer solution. After stirring at 70 °C for another 3 h, the solution was added by methanol (3 mL),

and further stirred for 30 min to ensure the consumption enough of isocyanate groups. Then, the solution was poured into a Poly tetra fluoroethylene (PTFE) plate that was subsequently put in a vacuum oven at 90 °C for 12 h to allow the reaction to complete. By dissolving the crude polymer in 40 mL of $CH_2Cl_2$ and then precipitating it into petroleum ether, we successfully obtained the purified P3 sample (16 g, yield 60%) as a yellow brown elastomer. The purification process was repeated three times. GPC using THF as the eluent measured the average molecular weight as $M_n = 63$ KDa and $M_w = 112$ KDa, and $M_w/M_n = 1.77$. The [1]H NMR spectrum of P3 is shown in Supplementary Fig. 2. [1]H NMR ($CDCl_3$, room temperature, 400 MHz) $\delta$ (ppm): 8.73–8.75 (d), 8.31–8.50 (s), 7.20–7.31 (d), 6.60–6.71 (d), 4.10–4.25 (t), 3.71 (t), 3.42 (t), 3.05 (m), 1.75 (m), 0.71–1.52 (m), 0.52 (t), 0–0.25 (s). The same procedure was used to prepare different polymers with different unit proportions, as summarized in the Supplementary Table 1.

**Film preparation for self-healing and mechanical property testing.** P3 (1.00 g) was dissolved in 6 mL of $CH_2Cl_2$. Subsequently, the solution was ultrasonicated to remove bubbles and poured into a PTFE dish (diameter 3.5 cm) that was covered with a glass petri dish, slowly evaporated at room temperature overnight. The resulting film was turned over and dried again for 8 h. Afterward, the film was dried at 65 °C in a vacuum oven for 12 h. After drying, the film was cut into small pieces for further testing. The same procedure was used to prepare films based on other ratios (PXs).

Mechanical tensile-stress experiments were performed using a Shanghai Hualong Electronic Universal Testing Machine equipped with a 200 N load cell at room temperature (20 °C). Unless otherwise specified, the deflection rate of uniaxial tensile measurements was 10 mm min$^{-1}$. Young's modulus was determined from the initial slope of the stress–strain curves. Rectangular shape specimens were used to the elastomer with dimension of $20 \times 14 \times 1$ mm$^3$. At least three specimens were tested and averaged for each sample. Cycle tensile tests were performed at a tensile rate of 10 mm·min$^{-1}$ and a recovery rate of 10 mm min$^{-1}$ at room temperature.

For the self-healing tests, the polymer films were cut into two pieces and put together to heal in different conditions. The healed polymer films were stretched following the same procedure to obtain the stress–strain curves. The mechanical healing efficiency, $\eta$, was defined as the ratio between the restored fracture strain relative to the original fracture strain.

**The conducting composite preparation for the stretchable electrode.** The concentrated P3/$CH_2Cl_2$ solution (0.1 g/mL) was poured into a PTFE cylindrical mold (inner diameter was 5 mm with a 2 mm inner solid column) and dried slowly at room temperature overnight, and dried at 70 °C in a vacuum oven for 8 h. The round polymer tube was prepared with 36 mm long, 1.5 mm thick, and 2 mm inside diameter. The P3 tube was then peeled off the PTFE cylindrical mold, and EGaIn liquid metal was applied as a conductive material to be infused in the tube by a syringe and needle. Conductive metal blocked in both directions of the tube, so EGaIn was completely encapsulated without leakage.

## Data availability

The source data underlying Figs. 2b–e, g, and 3c–e, Supplementary Figs. 4, 5, 6, 7, 8a–j, 9, 10a, b, 11, 12, 15, 16 and 17 are provided as a Source Data file. The data underlying all figures in the main text and supplementary information are publicly available at https://figshare.com/projects/source_data/77592.

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

## Acknowledgements

This research is supported by financial support from the Qingdao National Laboratory for Marine Science and Technology QNLM2016ORP0407, the National Natural Science Foundation of China (21621004, 21961132005, 21422605, 21908160), the Tianjin Natural Science Foundation, 18JCYBJC29500 and the China Postdoctoral Science Foundation 2019M651041.

## Author contributions

H.S.G., J.Y., and L.Z. conceived and designed the experiments. All the authors carried out the experiments. H.S.G., W.Q.Z., Y.H., and L.Z. performed the data and analysis. H.S.G, J.Y., and L.Z. wrote the manuscript. All the authors discussed the results and manuscript.

## Competing interests

The authors declare no competing interests.
