## [Peer Review File · Nature Communications]

Reviewers' comments:

Reviewer #1 (Remarks to the Author):

The paper describes a highly stretchable elastomer that can be stretched to 14,000%. It exhibits self-healing properties by using H-bonds and disulfide bonds. Healing experiments were demonstrated in various conditions e.g. underwater, at refrigerator, freezing temperature and in super-cooled high-concentrated saltwater. It also can be made into self-healing conductors by adding liquid metal, which is a common strategy used in literature. However, the presented material is not new and does not have desired properties as the authors suggested. Given the well-studied mechanism of healing using H-bonds and disulfide bonds, this paper does not represent a major advance sufficient for the journal.

Below are major issues that need to be addressed:

1. The title and abstract suggest that the polymer fabricated is an elastomer. However, the mechanical properties presented did not show elastomeric behavior. Is it a viscous material rather than elastic material (graphs in Fig 2)?

2. Is it a free-standing material that will not deform over time? It looks like a viscous material as shown in Fig 3d. The author did mention the need for strength in dynamic self-healing materials, but the material mechanical properties are low (moduli about 0.1 MPa) as indicated in the mechanical properties of the materials.

3. Do the materials mechanical properties stabilize over multi-cycles? It seems that at 3 cycle, the properties are still changing (Fig 2).

4. The properties of the materials are not that novel, and the self-healing mechanism is not new, e.g. Jian et al., *Polym. Adv. Technol.* 29:463 (2017). Underwater autonomous self-healing PDMS is not new (Kang et al. *Adv. Mater.* 30:1706846 (2018)). Autonomous self-healing polymer in extreme environments such as low temperature (basically rely on low T_g of materials, which exist in almost all autonomous healing polymers) and salt environment was also published before (Cao et al. *Nat Electron* 2:75 (2019)). Hence, it is not clear that what the authors presented constitutes a new direction as indicated in the abstract.

Reviewer #2 (Remarks to the Author):

The manuscript introduces a design and demonstration of a universally self-healing and highly stretchable elastomer, which can achieve fast self-healing under universal a variety of conditions, including under water, at room temperature, at freezing temperature (−20 °C) and in supercooled high-concentrated saltwater etc. At the same time, the material exhibits high stretchability up to 140 times. The authors attribute the self-healing properties to the incorporations of multi-strength hydrogen bonds and disulfide crosslinking sites in polydimethylsiloxane polymers. The high stretchability is assigned to the breakage and re-formation of intra/interchain hydrogen bonds and irreversible unfolding and refolding of polymer chains. Although the self-healing property of the material is rather competitive, we feel that the material design is quite similar to Bao's work (ref 4). They have obtained an elastomer with high toughness and good stretchability, which can also self-heal underwater and even in artificial sweat by incorporating hydrogen bonds into PDMS polymers. The authors may have a chance for publication on Nature Communications if the following major concerns are fully addressed:

- a) What's innovation parts compared with Bao's work (Ref 4)?
- b) The mechanism of the high stretchability is not clear. More evidence and data are needed to address the stretchability.
- c) The authors attribute the high fracture energy to the disulfide bonds. However, hydrogen bonds are more likely to play the major role. More control experiments are needed to clarify this point.
- b) No clear peaks can be found on the DSC curves.

Reviewer #3 (Remarks to the Author):

This is an interesting paper to deal with a challenging problem of self-healing in harsh conditions without external stimulus. To tackle this problem, the authors developed a novel elastomer based on a dynamic supramolecular polymer network. The dynamic network incorporated multi-type dynamic bonds, merging the unique properties of extremely large stretchability and rapid self-healing in ambient and harsh conditions, even in supercool high-concentrated saltwater. They found that the unique properties of elastomer were depended on the synergistic effects of different types of dynamic bonding reactions, and concluded that the strong hydrogen bonds and covalent disulfide metathesis mainly contributed to stretchability and self-healing ability. Overall, this elastomer system is new and well designed. Some concerns need to be addressed before publication.

1. From the paper, among three types of dynamic bonds, disulfide metathesis is the most critical for self-healing properties. From a mechanistic viewpoint, it is unclear how they contribute to underwater self-healing or anti-freezing self-healing properties?
2. As compared to many other metal-ligand coordination for self-healing materials, (Cheng-Hui L. Nat. Mater. DOI: 10.1038/NCHEM.2492; Ying-Li R. J. Am. Chem. Soc. DOI: 10.1021/jacs.6b02428), what are main advantages of this design?
3. When liquid metal EGaIn was used in the self-healing and stretchable conductor, is it possible that EGaIn will leak if the conductor is broken? How does it work when applied for underwater self-healing? Some potential electronic applications should be added to broaden the impact of this materials.

Dear editor:

Thank you very much for your email regarding our manuscript ‘Universally autonomous self-healing elastomer with high stretchability’ (No. NCOMMS-19-1125182-T). We do appreciate the efforts and valuable suggestions from the reviewers. The point-by-point replies and changes made in our revised manuscript according to the reviewers’ comments are listed below. All changes have been highlighted in red in the marked copy of the revised manuscript. Based on our careful revision and detailed supplementary information, we believe that the novelty and significance of this manuscript would be remarkable.

Reviewer #1:

[General comment]

The paper describes a highly stretchable elastomer that can be stretched to 14,000%. It exhibits self-healing properties by using H-bonds and disulfide bonds. Healing experiments were demonstrated in various conditions e.g. underwater, at refrigerator, freezing temperature and in super-cooled high-concentrated saltwater. It also can be made into self-healing conductors by adding liquid metal, which is a common strategy used in literature. However, the presented material is not new and does not have desired properties as the authors suggested.

Given the well-studied mechanism of healing using H-bonds and disulfide bonds, this paper does not represent a major advance sufficient for the journal.

[General response]

We do appreciate the enlightening comments from Reviewer#1! They are really useful for us to examine and improve our work from different perspectives. Aiming to represent the significance of this manuscript, we have added the detailed experiments, and compared the autonomous self-healing ability of this elastomer system with that of the elastomer in the mainly related literature in detail, as shown in **Table R1 (Figure 3F)**.

Table R1. The comparison of autonomous self-healing ability of this elastomer system and the elastomer in the related literature (best performance)

Key for self-healing	The condition or environment for autonomous self-healing (best performance)						Reference
	Underwater	Salt environment	Freezing temperature	Supercooled salt environment	Strong acid environment	Strong alkali environment	
Dopamine	√	√	×	×	×	×	1-4
Metal coordination bonds	×	×	-20°C	×	×	×	5-11
Strong H bonds	√	√	×	×	×	×	12, 13
Multi-strength H bonds	√	√	×	×	×	×	14-16
Dynamic borate/diol	√	×	×	×	×	pH=7-9	17
Dipole-dipole or ion-dipole interaction	√	√	×	×	pH=1.3 ~25% self-healing efficiency	pH=13 ~24% self-healing efficiency	18-20
Synergistic interaction (Strong and weak H bonds, disulfide metathesis)	√	√	-40°C	√	pH=0 ~84% self-healing efficiency	pH=14 ~88% self-healing efficiency	Our work

1. *Chem. Sci.* 2016, 7, 2736–2742
2. *Polym. Int.* 2019, 68, 1084–1090
3. *J. Mater. Chem. A.* 2018, 6, 6667–6674
4. *ACS Appl. Mater. Inter.* 2016, 8, 19047–190
5. *Nat. Chem.* 2016, 8, 618–624
6. *Adv. Mater.* 2019, DOI: 10.1002/adma.201901402
7. *Adv. Mater.* 2018, DOI: 10.1002/adma.201801435
8. *Nat Commun.* 2019, DOI:10.1038/s41467-019-09130-z
9. *Nat Commun.* 2019, DOI: 10.1038/s41467-018-05285-3
10. *J. Am. Chem. Soc.* 2016, 138, 6020–6027
11. *Adv. Funct. Mater.* 2019, 29, 1901058
12. *Angew. Chem. Int. Ed.* 2018, 57, 11242.
13. *J. Am. Chem. Soc.* 2018, 140, 15, 5280-5289
14. *Adv. Funct. Mater.* 2019, DOI: 10.1002/adfm.201907109
15. *Angew. Chem. Int. Ed.* 2017, 56, 8795.
16. *Adv. Mater.* 2018, DOI: 10.1002/adma.201706846
17. *J. Mater. Chem. A,* 2016, 4, 14122–14131
18. *Adv. Mater.* 2018, DOI: 10.1002/adma.201804602
19. *Nat. Electron.* 2019, 2, 75–82,
20. *Adv. Mater.* 2017, 29, 1605099

From this table, the advance of this elastomer system based on synergistic interaction is well presented:

1. Significance in material performance: This elastomer not only shows high stretchability (14,000%), but also can autonomous heal itself in various harsh

conditions, including underwater (93% efficiency), at freezing and ultralow temperatures (-20 °C and -40 °C), in the supercooled high-concentrated saltwater (30% NaCl solution at -10 °C, 89% efficiency), and in strong acid/alkali environment (pH=0 or 14, 88% or 84% efficiency). Actually, the development of synthetic materials that can be autonomous self-healing in each of above-mentioned harsh conditions remains difficult to address. To the best of our knowledge, this is the first report of autonomous self-healing in such universal conditions without external stimulus, as well as achieving the high self-healing efficiency.

2. Significance in material design: The universally self-healing capability of this elastomer is mainly contributed by synergistic interaction among the dynamic strong H bonds, weak H bonds, and disulfide exchange, as shown in **Fig.1**. The self-healing efficiency of this elastomer based on the synergistic interaction is significantly greater than the simple sum efficiencies of a material based on two types of the dynamic bonds, as presented in **Fig.3E**. What's more, in the basis of the systematic control tests of 'parent' materials, we found that strong-H bonds are highly critical for stretchability and disulfide metathesis is the most important to universally self-healing property.

This elastomer would be highly important for applications of self-healing electronic equipment (e.g. electronic detection sensors or communication devices) in marine area, polar regions, and industrial wastewater, etc.

The details of point-by-point replies were provided as below:

[Comment 1.1] The title and abstract suggest that the polymer fabricated is an elastomer. However, the mechanical properties presented did not show elastomeric behavior. Is it a viscous material rather than elastic material (graphs in Fig 2)?

[Response 1.1]

Thank you for pointing this out! Actually, the materials in this work (PDMS–SS–IP–BNB) presented the mechanical property between an ideal elastic material behavior (in accord with Hooke's law) and ideal viscous material behavior (in accord with Newton's law). It depended on the stretching speed and unit mass ratio. As shown in **Fig.2B** and **C**, PDMS–SS–IP–BNB showed a representative elastomeric behavior for P6 and P3 at high stretching speed. In many previous literatures, the materials that presented this property were identified 'elastomer':

1. *Nat. Chem.* 2016, DOI: 10.1038/NCHEM.2492
2. *Adv. Mater.*, 2018, DOI: 10.1002/adma.201706846
3. *Macromol. Rapid Commun.*, 2017, DOI: 10.1002/marc.201700110
4. *Macromol. Rapid Commun.*, 2018, DOI: 10.1002/marc.201800349
5. *Macromol. Chem. Phys.*, 2017, DOI: 10.1002/macp.201700409
6. *J. Mater. Chem. A*, 2019, DOI: 10.1039/c9ta09158k
7. *Macromol. Rapid Commun.*, 2016, DOI: 10.1002/marc.201600300
8. *Polymer*, 2015, DOI: org/10.1016/j.polymer.2015.03.023
9. *Macromol. Rapid Commun.* 2017, DOI: 10.1002/marc.201700686
10. *Adv. Funct. Mater.*, 2018, DOI: 10.1002/adfm.201800741
11. *J. Mater. Chem. A*, 2018, DOI: 10.1039/c7ta09841c

12. *J. Mater. Chem. B*, 2019, DOI: 10.1039/c9tb00831d

[**Comment 1. 2**] Is it a free-standing material that will not deform over time? It looks like a viscous material as shown in Fig 3d. The author did mention the need for strength in dynamic self-healing materials, but the material mechanical properties are low (moduli about 0.1 MPa) as indicated in the mechanical properties of the materials.

[**Response 1. 2**]

Negligible deformation of PDMS–SS–IP–BNB film was presented when the temperature was below 15 °C. When temperature was higher than 15 °C but lower than 30 °C, this material would deform to a certain extent after 48 h.

As is well known, it is a compromise between self-healing capability and material mechanical strength. In **Fig.3A**, the post-healing PDMS–SS–IP–BNB film was able to withstand 526 times of its own weight, indicating its favorable mechanical strength. Moreover, PDMS–IP–SS–BNB film showed the notch-insensitive ability in **Fig.2F-G**, and the notch was blunted without tearing further during the stretching process, thus evidencing the extraordinary toughness of PDMS–IP–SS–BNB material. In addition to universally self-healing capability with high healing efficiency, the mechanical properties of PDMS–SS–IP–BNB represented a superiority.

This phenomenon is also consistent with the previous reported self-healing materials:

1. *Adv. Mater.*, 2018, DOI: 10.1002/adma.201706846
2. *Adv. Mater.* 2018, DOI: 10.1002/adma.201804602
3. *Nat. Chem.* 2016, DOI: 10.1038/NCHEM.2492
4. *Adv. Mater.* 2017, DOI: 10.1002/adma.201605099

[**Comment 1. 3**] Does the materials mechanical properties stabilize over multi-cycles? It seems that at 3 cycle, the properties are still changing (Fig 2).

[**Response 1. 3**]

In **Fig.2E**, the cyclic stress-strain curves showed a pronounced hysteresis due to the energy dissipation during the rupture of the dynamic bonds. It cannot be accounted for the instability of the material. In **Fig. S11**, the stress-strain test of long-term stored PDMS-SS-IP-BNB was also evaluated to demonstrate its stable property.

The hysteresis is a representative phenomenon of dynamic bond-based materials as previously reported:

1. *Adv. Mater.* 2019, DOI: 10.1002/adma.201901402,
2. *Adv. Mater.* 2016, DOI: 10.1002/adma.201605099,
3. *J. Am. Chem. Soc.*, 2018, DOI: 10.1021/jacs.8b01682
4. *Adv. Mater.* 2018, DOI: 10.1002/adma.201801435
5. *Nat. Chem.* 2016, DOI: 10.1038/NCHEM.2492

[**Comment 1. 4**] The properties of the materials are not that novel, and the self-healing mechanism is not new, e.g. Jian et al., *Polym. Adv. Technol.* 29:463 (2017). Underwater autonomous self-healing PDMS is not new (Kang et al. *Adv. Mater.* 30:1706846 (2018)). Autonomous self-healing polymer in extreme environments such as low temperature (basically rely on low T_g of materials, which exist in almost all autonomous healing polymers) and salt environment was also published before (Cao et al. *Nat Electron* 2:75 (2019)). Hence, it is not clear that what the authors presented constitutes a new direction as indicated in the abstract.

[**Response 1. 4**]

Thank you for your comments!

To the best of our knowledge, this is the first report of high-stretchable and autonomous self-healing materials that enable healing in such universal conditions (underwater, freezing temperature -20°C, ultralow temperature -40°C, supercooled high-concentrated saltwater, and strong acid/alkali environment) without external stimulus, as well as achieving the high self-healing efficiency, as shown in **Fig. 3F**.

Figure 3F. A comparison between this work and previously-reported self-healing materials in terms of healing ability in universal harsh conditions

Actually, it is difficult to develop a healing material that is able to self-heal under one of harsh conditions: **underwater, freezing temperature, or acid/alkali environment**. The reasons are mainly as follows:

i) When healable materials are injured or fractured underwater, water molecules can disturb the reconnection of dynamic bonds, resulting in failure of the material to heal. (Underwater)

ii) Under freezing conditions, the dynamic character of bonds in healable materials can be significantly resisted, thus extremely restraining the self-healing process. Lower temperatures, more resistance. (Freezing/ultralow temperatures)

iii) Some “keys” for self-healing such as metal coordination bonds and dopamine are susceptible to pH changes. (Acid/alkali environments)

However, these harsh conditions commonly cannot be avoided due to the requirement of self-healing material applied in marine area, polar regions, or industrial wastewater.

Therefore, it is great challenging and desirable to develop a universally self-healing material in all these harsh conditions, as well as achieving high efficiency.

As the reviewer mentioned, the paper (Kang et al. *Adv. Mater.* 2018,30:1706846) described a water-insensitive and ambient self-healing material (r.t. for 48 h, healing efficiency 78%) based on strong and weak crosslinking H-bonds. In this work, we focused on the development of universally self-healing materials based on synergistic interaction, as well as achieving high healing efficiency. Notably, our work designed the ‘parent material’ P5 based on the strong and weak H-bonds, and its self-healing capability was negligible in freezing (-20°C) or ultralow temperature (-40°C) or strong acid/alkali environment (**Fig. 3E**). This result demonstrated the significance of our material that was based on the synergistic interaction, and also explained that anti-freezing self-healing ability not just relied on low T_g of the materials.

The paper (Cao et al. *Nat Electron.* 2019, 2:75) developed a self-healing material based on ion-dipole interaction, capable of healing under seawater and acid/alkali environment (pH=1.3 or 13). But its self-healing efficiency is 22.6% in r.t., ~30% in seawater, 20.9% in pH 1.3 solution, and 17.4% in pH 13 solution. What’s more, the anti-freezing self-healing property was not mentioned in the paper.

Overall, based on the synergistic interaction, the material in our work can self-heal under all these harsh conditions with high healing efficiency and super-strechability. We believe that it has presented a significant advance and novelty.

Figure 3E. The self-healing efficiency (%) of P1 (based on S-S and weak H-bonds), P3 (PDMS–SS–IP–BNB) and P5 (based on strong and weak H-bonds) film healed underwater for 12 h, at -20°C for 12 h, at -40°C for 24 h, 30% NaCl solution at -10 °C for 24h, pH=0 and pH=14 solution for 24 h.

Reviewer #2:

The manuscript introduces a design and demonstration of a universally self-healing and highly stretchable elastomer, which can achieve fast self-healing under universal a variety of conditions, including under water, at room temperature, at freezing temperature ($-20\text{ }^{\circ}\text{C}$) and in supercooled high-concentrated saltwater etc. At the same time, the material exhibits high stretchability up to 140 times. The authors attribute the self-healing properties to the incorporations of multi-strength hydrogen bonds and disulfide crosslinking sites in polydimethylsiloxane polymers. The high stretchability is assigned to the breakage and re-formation of intra/interchain hydrogen bonds and irreversible unfolding and refolding of polymer chains. Although the self-healing property of the material is rather competitive, we feel that the material design is quite similar to Bao's work (ref 4). They have obtained an elastomer with high toughness and good stretchability, which can also self-heal underwater and even in artificial sweat by incorporating hydrogen bonds into PDMS polymers. The authors may have a chance for publication on Nature Communications if the following major concerns are fully addressed:

[Comment 2. 1] What's innovation parts compared with Bao's work (Ref 4)?

[Response 2. 1]

Thank you very much for these valuable comments! They are highly useful to improve our work.

Bao's group presented an excellent work based on strong and weak H-bonds, focused on the water-insensitive self-healing elastomer with 1200% of stretchability. In this work, we represented a 14000% stretchability of elastomer based on synergistic interaction, focused on the development of a novel material that is able to universally self-healing in all of harsh conditions, including supercooled high-concentrated saltwater, freezing or even ultralow temperature, and strong acid/alkali environment, as well as achieving high healing efficiency (**Fig. 3**).

Actually, the 'parent' material (P5) in this work basing on the strong and weak H-bonds was not able to heal itself in freezing ($-20\text{ }^{\circ}\text{C}$) or ultralow temperature ($-40\text{ }^{\circ}\text{C}$), in $-10\text{ }^{\circ}\text{C}$ of saltwater, and in strong acid/alkali environment (**Fig.3E**).

[Comment 2. 2] The mechanism of the high stretchability is not clear. More evidence and data are needed to address the stretchability.

[Response 2. 2]

Thank you for your pointing it out! We have added more discussions in the revised manuscript.

[Comment 2. 3] The authors attribute the high fracture energy to the disulfide bonds. However, hydrogen bonds are more likely to play the major role. More control experiments are needed to clarify this point.

[Response 2. 3]

This suggestion is highly useful! We have added more control experiments and discussions in the revised manuscript.

As shown in **Fig. 2B** and **Table S2**, P2 presents a lower stretchability (100%); the

lack of BNB units allowed for strong H-bonds of BNB–BNB crosslinks. The strain at break of the P5 film (lack of SS) was ca. 2,000%, higher than that of P2 but still much lower than that of P4 (14,000%). Based on the results, we can propose that the high fracture energy and stretchability of elastomer is contributable to synergistic interaction among the dynamic strong H-bonds, weak H-bonds, and disulfide metathesis in a supramolecular polymer network, and the strong H-bonds play the most important role.

[Comment 2. 4] No clear peaks can be found on the DSC curves.

[Response 2. 4]

This phenomenon is consistent with previous reports, as presented in **Fig. R1** as an example (*Nat Commun*, 2019, DOI:10.1038/s41467-019-09130-z, *Adv. Mater.* 2018, DOI: 10.1002/adma.201706846). It is possibly because that PDMS–SS–IP–BNB elastomers possess smaller thermal effects, which also benefit for anti-freezing self-healing ability.

Figure R1. Differential scanning calorimetry (DSC) thermal analysis of PDMS-MPU_{0.4}-IU_{0.6} (red), PDMS-MPU_{0.3}-IU_{0.7} (orange) and PDMS-MPU_{0.2}-IU_{0.8} (blue) (*Adv. Mater.* 2018, DOI: 10.1002/adma.201706846)

Reviewer #3:

This is an interesting paper to deal with a challenging problem of self-healing in harsh conditions without external stimulus. To tackle this problem, the authors developed a novel elastomer based on a dynamic supramolecular polymer network. The dynamic network incorporated multi-type dynamic bonds, merging the unique properties of extremely large stretchability and rapid self-healing in ambient and harsh conditions, even in supercool high-concentrated saltwater. They found that the unique properties of elastomer were depended on the synergistic effects of different types of dynamic bonding reactions, and concluded that the strong hydrogen bonds and covalent disulfide metathesis mainly contributed to stretchability and self-healing ability. Overall, this elastomer system is new and well designed. Some concerns need to be addressed before publication.

[Comment 3. 1] From the paper, among three types of dynamic bonds, disulfide metathesis is the most critical for self-healing properties. From a mechanistic viewpoint, it is unclear how they contribute to underwater self-healing or anti-freezing self-healing properties?

[Response 3. 1]

Thank you for your positive and valuable comments!

In **Fig.3E**, we have presented the underwater, anti-freezing, and anti-acid/alkali self-healing efficiencies of P5 material, which was lack of S-S bonds. The results indicated that P5 was not able to heal itself at -20°C, -40°C, -10°C saltwater and in strong acid/alkali environment. Because compared to most covalent bonds, the lower S-S bond dissociation energy (60 kcal/mol) and disulfide metathesis reaction efficiently improved the healing performance. We have added more discussions in the revised manuscript.

[Comment 3. 2] As compared to many other metal-ligand coordination for self-healing materials, (Cheng-Hui L. Nat. Mater. DOI: 10.1038/NCHEM.2492; Ying-Li R. J. Am. Chem. Soc. DOI: 10.1021/jacs.6b02428), what are main advantages of this design?

[Response 3. 2]

Metal-ligand coordination is susceptible to pH changes. Therefore, the materials based on these dynamic bonds would be unstable in either or both strong acid and alkali environment. As far as we know, this kind of materials cannot achieve universally self-healing property (**Fig.3F**).

[Comment 3. 3] When liquid metal EGaIn was used in the self-healing and stretchable conductor, is it possible that EGaIn will leak if the conductor is broken? How does it work when applied for underwater self-healing? Some potential electronic applications should be added to broaden the impact of this materials.

[Response 3. 3]

Thank you for your pointing this out!

When the conductor was broken, the surface tension and poor fluidity of EGaIn should inhibit it leaking and the cut material would touch each other as soon as possible.

When applied in water, the hydrophobicity of polymer backbone (PDMS) induces an ordered arrangement of the surrounding water molecules, reducing the impact of the inner conductor.

We have also proposed some electronic applications of this elastomer, such as recoverable liquid metal flexible sensors. More perspectives have been added in the revised manuscript.

Thank you very much!

Sincerely,

Lei Zhang, Ph.D.

Professor and Chair

Department of Biochemical Engineering

School of Chemical Engineering and Technology

Tianjin University, Tianjin, China

Email: lei_zhang@tju.edu.cn

Cell: 86-13502038015

Reviewers' comments:

Reviewer #3 (Remarks to the Author):

The authors have addressed my and other comments in a very persuasive way. I recommend it for publication.

Reviewer #4 (Remarks to the Author):

The authors have answered the questions that reviewers suggested. From my point of view, I think molecular design is reasonable, the performance of the material is impressive. Moreover, it gives us a concept of universally autonomous self-healing system which combines three different kinds of dynamic interactions to overcome the autonomous self-healing especially under cold environment and acid/alkali conditions. It may be acceptable after addressing the following issues:

1. Line 165-166: In temperature ramp experiment, the solid to liquid transition should be explained more carefully.
2. The explanation of the high stretchability should add the chain conformations of PDMS soft segments, this also contribute to the high stretchability of the system. And What is the phase behavior of the system, is there microphase separation that also contribute to the stretchability? The author should take these factors into consider.
3. Line 230-232. During the stretch, the hydrogen bonds break and reform will be the major contribution to high stretchability. Can the authors explain the contribution of disulfide metathesis in the time scale of extension experiment?
4. Besides the flexibility of polymer backbone, the role of dynamic bonds for self-healing in freezing conditions should be explained.

Reviewer #5 (Remarks to the Author):

I do not think the authors have successfully addressed Reviewers #1 and #2's comments.

Reviewer #1's concern was that "...the presented material is not new and does not have desired properties as the authors suggested."

It is counterintuitive for an elastomer to be stretched by 140 times without break. The tensile tests were not performed using standard dog-bone shape samples. Therefore, the claimed stretchability is highly suspicious.

Based on data reported, the material is more like a viscoelastic liquid, but not an elastomer.

The concept of using reversible bonds with multiple levels of strength to achieve self-healing ability is not new.

Reviewer #2's concern was on the novelty of the concept and the scientific understanding of the self-healing mechanism. Neither of the two was appropriately addressed in the revision.

The authors have addressed my and other comments in a very persuasive way. I recommend it for publication.

[General Response]: Thanks very much for your kind comments!

Comments of Reviewer #4:

The authors have answered the questions that reviewers suggested. From my point of view, I think molecular design is reasonable, the performance of the material is impressive. Moreover, it gives us a concept of universally autonomous self-healing system which combines three different kinds of dynamic interactions to overcome the autonomous self-healing especially under cold environment and acid/alkali conditions. It may be acceptable after addressing the following issues:

[Comment 4.1] Line 165-166: In temperature ramp experiment, the solid to liquid transition should be explained more carefully.

[Response 4.1] Thanks very much for your positive comments and valuable suggestions! We have added more discussions according to this comment in the revised manuscript (2.3 Rheological and Mechanical Properties).

[Comment 4.2] The explanation of the high stretchability should add the chain conformations of PDMS soft segments, this also contribute to the high stretchability of the system. And What is the phase behavior of the system, is there microphase separation that also contribute to the stretchability? The author should take these factors into consider.

[Response 4.2] Thank you for pointing this out! We have added the contribution of PDMS soft segments to stretchability in the revised manuscript.

In this work, there is microphase separation induced by the micro-scale aggregation of hard segment via strong H bonding force between BNB. Therefore, the main contribution of micro-phase separation is actually the dynamic strong H-bonds to elastomer stretchability, which has been further explained in the revised manuscript (*Nat. Commun.*, 2019, <https://doi.org/10.1038/s41467-019-10061-y>, *Ind. Eng. Chem. Res.*, 2020, <https://dx.doi.org/10.1021/acs.iecr.9b06107>).

[Comment 4.3] Line 230-232. During the stretch, the hydrogen bonds break and reform will be the major contribution to high stretchability. Can the authors explain the contribution of disulfide metathesis in the time scale of extension experiment?

[Response 4.3] During the stretch, the weaker aromatic disulfide metathesis can act as the sacrificial bonds for breakage, which can dissipate the strain energy to prevent the rupture of main polymeric backbones, thus enhancing the stretchability of this elastomer (*Macromol. Rapid Commun.* 2018, 1700686 DOI: 10.1002/marc.201700686).

[Comment 4.4] Besides the flexibility of polymer backbone, the role of dynamic bonds for self-healing in freezing conditions should be explained.

[Response 4.4] In freezing conditions, the sufficient multiple dynamic bonds can synergistically provide more reconfiguration opportunities when polymer chains re-entangle at the damaged surface. More importantly, the low bond energy of disulfide metathesis can further ensure the easy reconnection in freezing conditions. They are the contributions of dynamic bonds to anti-freezing self-healing property (*Chem. Mater.* 2018, 30, 6026–6039). We have added more discussions in the revised manuscript.

Comments of Reviewer #5:

I do not think the authors have successfully addressed Reviewers #1 and #2's

comments.

Reviewer #1's concern was that "...the presented material is not new and does not have desired properties as the authors suggested."

It is counterintuitive for an elastomer to be stretched by 140 times without break. The tensile tests were not performed using standard dog-bone shape samples. Therefore, the claimed stretchability is highly suspicious.

Based on data reported, the material is more like a viscoelastic liquid, but not an elastomer.

The concept of using reversible bonds with multiple levels of strength to achieve self-healing ability is not new.

Reviewer #2's concern was on the novelty of the concept and the scientific understanding of the self-healing mechanism. Neither of the two was appropriately addressed in the revision.

[General Response] As requested by the editor, we summarize the comments of Reviewer #5 as the following two points:

[Comment 5.1] It is counterintuitive for an elastomer to be stretched by 140 times without break. The tensile tests were not performed using standard dog-bone shape samples. Therefore, the claimed stretchability is highly suspicious.

[Response 5.1] Thank you for your comments!

To the best of our knowledge, rectangular shape specimens are most commonly used in the tensile tests of elastomer materials, as reported in most related literatures in this field [1-19]. Because elastomers are stretchable and elastic (up to 14,000% of stretchability in this work), thus it's not necessary to prepare them into a dog-bone shape, as shown in **Fig. R1**.

However, dog-bone shape becomes necessary when the materials are more plastic, meaning they are more brittle and their stretchability is usually below 500% [20]. The dog-bone shape design can help because the transitional circular areas in this shape as shown in **Fig. R1** are specifically designed to significantly reduce the stress concentration to rupture at the grip sections, thus avoid the failure of tensile test [21-26].

Therefore, it is clear that all the tests used in this work are standard and up to date. We have added more details on the tensile tests in the revised supporting information.

Fig. R1 Schematic diagram of dog-bone shape specimen for plastics in tensile test.

Reference:

1. *Chem. Mater.* 2019, DOI: 10.1021/acs.chemmater.9b02136
2. *Adv. Funct. Mater.* 2019, 29, 1901058, DOI: 10.1002/adfm.201901058
3. *Adv. Mater.* 2019, 1901402, DOI: 10.1002/adma.201901402
4. *Adv. Mater.* 2019, 1904029, DOI: 10.1002/adma.201904029
5. *Nat Electron*, 2019, 75, DOI: org/10.1038/s41928-019-0206-5
6. *ACS Appl. Mater. Interfaces* 2019, 18720, DOI: 10.1021/acsami.9b03346
7. *Adv. Mater.* 2018, 1706846, DOI: 10.1002/adma.201706846
8. *Macromol. Rapid Commun.* 2016, DOI: 10.1002/marc.201600300
9. *Chem. Eng. J.* 2019, 121993, DOI: org/10.1016/j.cej.2019.121993
10. *Nat. Chem.* 2016, DOI: 10.1038/NCHEM.2492
11. *Macromol. Rapid Commun.* 2018, 1700686, DOI: 10.1002/marc.201700686
12. *J. Am. Chem. Soc.* 2016, 6020, DOI: 10.1021/jacs.6b02428
13. *Nat Commun*, 2019, 1164, DOI: org/10.1038/s41467-019-09130-z
14. *Nat Commun*, 2019, 2158, DOI: org/10.1038/s41467-019-10061-y
15. *Adv. Mater.* 2018, 1801435, DOI: 10.1002/adma.201801435
16. *Chem. Commun.*, 2017, 12088, DOI: 10.1039/c7cc06126a
17. *Adv. Mater.* 2017, 1705145, DOI: 10.1002/adma.201705145
18. *Adv. Mater.* 2018, 1804602, DOI: 10.1002/adma.201804602
19. *Adv. Mater.* 2018, 1706846, DOI: 10.1002/adma.201706846

20. Standard Test Method for Tensile properties of Plastics, ASTM D638-03
21. *Eng. Fract. Mech.* 2019, DOI: org/10.1016/j.engfracmech.2019.02.027
22. *Int. J. Fract.*, 1965, DOI: org/10.1007/BF00186856
23. *ACS Nano*, 2017, 6271, DOI: 10.1021/acsnano.7b02493
24. *J. Wood, Sci*, 2014, 287, DOI: org/10.1007/s10086-014-1398-y
25. *Mech. Adv. Mater. Struc.*, 2004, 51, DOI: 10.1080/15376490490257666
26. *Prog. Addit. Manuf.*, 2016, 21, DOI: org/10.1007/s40964-015-0002-3

[**Comment 5.2**] Based on data reported, the material is more like a viscoelastic liquid, but not an elastomer.

[**Response 5.2**] Based on our results, the material developed in this work was identified as an ‘elastomer’ according to the following reasons:

(1) According to IUPAC definition, an elastomer is ‘a polymer with rubber-like elasticity’, which means it possesses the capacity to sustain a large deformation without rupture and recover spontaneously (<Principles of Polymer Chemistry>, 1992, P432). In this work, as shown in **Fig. 2E**, the cyclic stress-strain test presents the stretch-recovery curves for multi-cycles, clearly proving that our material possesses the large deformation and spontaneous recovery ability, the characteristic of elastomer, consistent with many literature reports [1-10].

Fig. 2E) The stress–strain curve of our material in cyclic stress–strain tests (30% strain) in successive stretching.

(2) Moreover, the stress-strain curve of this material presents a typical shape of the elastomer that possesses interchain momentary bonds, as shown in **Fig. R2B**. The interchain momentary bonding results in a net increase of stiffness above the elastic curve in extension, illustrating an envelope curve that has been well documented, e.g. **Fig. R2A**. (<Fatigue, Stress, and Strain of Rubber Components>, 2008, P19-41)

Fig. R2 (A) A typical stress-strain curve of elastomer in the presence of interchain momentary bonds documented in the book and **(B)** the stress-strain curve of our material

Viscoelastic liquids present a flow state (such as asphalt) and their loss modulus (G'') is greater than storage modulus (G') [11-15]. As shown in **Fig. S8B**, the rheological test results of our material show obvious $G' > G''$, a characteristic feature of elastomer, consistent with many literature reports [16-22].

Figure S8B Storage modulus (G') and loss modulus (G'') of our material versus frequency at room temperature.

Reference:

1. *Adv. Mater.* 2018, 30, 1804602, DOI: 10.1002/adma.201804602
2. *Adv. Mater.* 2018, 30, 1706846, DOI: 10.1002/adma.201706846
3. *Nat. Chem.* 2016, DOI: 10.1038/NCHEM.2492
4. *Adv. Funct. Mater.*, 2018, DOI: 10.1002/adfm.201800741
5. *Macromol. Rapid Commun.* 2018, 39, 1700686, DOI: 10.1002/marc.201700686
6. *Adv. Funct. Mater.* 2019, 1907109, DOI: 10.1002/adfm.201907109
7. *Adv. Mater.* 2019, 1901402, DOI: 10.1002/adma.201901402
8. *Adv. Mater.* 2018, 1801435, DOI: 10.1002/adma.201801435
9. *Chem. Eng. J.* 2019, 121993, DOI: org/10.1016/j.cej.2019.121993
10. *Macromolecules* 2019, 4209, DOI: 10.1021/acs.macromol.9b00128
11. *Sci. Adv.*, 2018, 6243, DOI: 10.1126/sciadv.aao6243
12. *Dent. Mater.*, 2015, 1003, DOI: org/10.1016/j.dental.2015.05.009
13. *Polym. Test.*, 2016, DOI.org/10.1016/j.polymertesting.2016.08.001
14. *Soft Matter*, 2015, DOI: 10.1039/c5sm00131e
15. *Macromolecules* 1995, 28, 512, DOI: org/10.1021/ma00106a014
16. *Chem. Mater.* 2018, 3752, DOI: org/10.1021/acs.chemmater.8b00832
17. *J. Mater. Chem. A*, 2018, 5887, DOI: 10.1039/c7ta09841c
18. *Polymer*, 2017, 189, DOI: org/10.1016/j.polymer.2017.05.060
19. *ACS Macro Lett.* 2016, 1196, DOI:10.1021/acsmacrolett.6b00662
20. *J.Mater.Chem.A*, 2017, 25660, DOI: 10.1039/c7ta08255j
21. *Adv. Mater.* 2017, 1702616, DOI: 10.1002/adma.201702616
22. *Adv. Mater.* 2018, 1802556, DOI: 10.1002/adma.201802556

REVIEWERS' COMMENTS:

Reviewer #4 (Remarks to the Author):

The authors have addressed all my questions.